# Mechanisms of Cisplatin Resistance in HPV Negative Head and Neck Squamous Cell Carcinomas

**DOI:** 10.3390/cells11030561

**Published:** 2022-02-05

**Authors:** Ana Belén Griso, Lucía Acero-Riaguas, Beatriz Castelo, José Luis Cebrián-Carretero, Ana Sastre-Perona

**Affiliations:** 1Laboratory of Experimental Therapies and Biomarkers in Cancer, IdiPAZ, 28046 Madrid, Spain; anabgriso98@gmail.com (A.B.G.); luciacero_97@hotmail.com (L.A.-R.); 2Medical Oncology Department, University Hospital La Paz, 28046 Madrid, Spain; castelobeatriz@gmail.com; 3Oral and Maxillofacial Surgery Department, University Hospital La Paz, 28046 Madrid, Spain; josel.cebrian@salud.madrid.org

**Keywords:** HNSCC, cisplatin, tumor microenvironment, cell plasticity, epigenetics, cancer stem cells

## Abstract

Head and neck squamous cell carcinomas (HNSCCs) are the eighth most common cancers worldwide. While promising new therapies are emerging, cisplatin-based chemotherapy remains the gold standard for advanced HNSCCs, although most of the patients relapse due to the development of resistance. This review aims to condense the different mechanisms involved in the development of cisplatin resistance in HNSCCs and highlight future perspectives intended to overcome its related complications. Classical resistance mechanisms include drug import and export, DNA repair and oxidative stress control. Emerging research identified the prevalence of these mechanisms in populations of cancer stem cells (CSC), which are the cells mainly contributing to cisplatin resistance. The use of old and new CSC markers has enabled the identification of the characteristics within HNSCC CSCs predisposing them to treatment resistance, such as cell quiescence, increased self-renewal capacity, low reactive oxygen species levels or the acquisition of epithelial to mesenchymal transcriptional programs. In the present review, we will discuss how cell intrinsic and extrinsic cues alter the phenotype of CSCs and how they influence resistance to cisplatin treatment. In addition, we will assess how the stromal composition and the tumor microenvironment affect drug resistance and the acquisition of CSCs’ characteristics through a complex interplay between extracellular matrix content as well as immune and non-immune cell characteristics. Finally, we will describe how alterations in epigenetic modifiers or other signaling pathways can alter tumor behavior and cell plasticity to induce chemotherapy resistance. The data generated in recent years open up a wide range of promising strategies to optimize cisplatin therapy, with the potential to personalize HNSCC patient treatment strategies.

## 1. Head and Neck Squamous Cell Carcinomas

Head and neck squamous cell carcinomas (HNSCCs) were the eighth most frequent tumors worldwide in 2020, with 931,931 newly diagnosed cases and 467,125 deaths [1]. HNSCC is a heterogenous disease that arises from the mucosal epithelium of the oral cavity, pharynx, larynx, nose, and salivary glands, with the former three being the most frequent [2]. Known causes include exposure to tobacco, excessive alcohol consumption, and infection with high-risk human papilloma virus strains (HPV16 and 18) [3]. 

HPV-positive and negative cancers have been considered two different clinical entities by the World Health Organization (WHO) since 2017 [4]. For this reason, HNSCCs are classified as HPV-positive or HPV-negative cancers. Further classification based on their location does not affect diagnosis. While squamous cell carcinomas (SCCs) from the oral cavity, the pharynx and the larynx are more correlated with smoking/alcohol (90–95%) [5], SCCs from the oropharynx are more commonly associated with HPV infection [6]. The rate of HPV-associated oropharyngeal squamous cell carcinoma has increased dramatically, and this has substantially altered their epidemiology. These tumors define a distinct subset of patients that have frequent lymph node involvement and an improved prognosis compared with HPV-negative, tobacco-driven oropharyngeal cancers. Furthermore, the use of an HPV vaccine that can prevent the infection of high-risk HPV oncogenic variants, including the ones causing HNSCCs, promises to decrease the number of these malignancies [7,8].

Improving the management of HPV-negative HNSCCs is an important unmet medical need. One of the main characteristics of these entities is the tumor mutational burden. Due to their link to smoking and alcohol consumption, HPV-negative tumors harbor a high mutational load as well as copy number alterations [9], and may be related to the poor outcome of these patients. For these reasons, despite early diagnosis and successful surgeries with clear margins, more than 10–20% of patients with early-stage cancers and more than 50% with advanced HPV-negative HNSCCs will develop locoregional or distal relapses, with a poor prognosis and overall survival (OS) rates of less than one year [2]. Systemic therapy containing cisplatin has been, for many years, the main choice to treat advanced HPV-negative HNSCCs. However, a broad issue has been the partial or null response to this drug [10] and the development of cisplatin refractory relapses that is associated with a very poor outcome, as we will discuss in the following sections. In this review, we will discuss known mechanisms of cisplatin resistance, focusing exclusively on HPV-negative cancers, since they are the main recipients of this treatment. We will discuss the most studied mechanisms, which implicate alterations in intracellular drug accumulation and detoxification and DNA damage repair on cancer cells. Furthermore, we will review other novel mechanisms of cisplatin resistance including the presence of cancer stem cells that show innate drug resistance, epigenetic changes that regulate cell plasticity and stem-like properties, and the involvement of the tumor microenvironment (TME).

### 1.1. Molecular Alterations of HVP Negative HNSCCs

HNSCCs present multiple genetic alterations that affect cell proliferation and differentiation, and the main genetic drivers are summarized in Table 1. They include inactivating mutations of the tumor suppressor *TP53* (altered in 84% of the patients) and loss of function and deletion of *CDKN2A* (p16/p14, 58%), which both contribute to increased cell proliferation and apoptosis evasion [9]. In some patients, the inhibition of apoptosis is driven by the inactivation of the cell death-related gene CASP8. Gain of function mutations or amplifications affecting classical oncogenes such as *PIK3CA* (34%), *MYC* (14%), and *CCND1* (31%) often result in increased cell proliferation and survival cues. Gain of function mutations and amplifications additionally occur in multiple receptor tyrosine kinases such as EGFR, *FGFR1*, and *IGFR*, converging on the activation of PI3K and MAPK signaling pathways, again boosting tumor proliferation and survival. 

Increased self-renewal of cancer cells is caused by commonly found copy number amplifications of the transcription factors (TFs) *TP63* and *SOX2* [11,12,13], which promote survival and inhibit squamous differentiation [14]. Self-renewal can also be enhanced by direct inhibition of cell differentiation pathways, such as mutations that result in the inhibition of the NOTCH pathway [15]. Stemness is also promoted by the loss of *FAT1* cadherin (25–30%) [16], which activates the Hippo pathway and guides tumor progression. Finally, increased cellular detoxification is driven by alterations in the NRF2/KEAP1 pathway [17].

**Table 1 cells-11-00561-t001:** Altered genes or pathways presented in HNSCC and their cellular effects.

Affected Gene/Pathway	Type of Alteration	Frequency	Effect on Cell	Ref.
**TP53**	Inactivation	84%	Increased cell proliferation and apoptosis evasion	[9]
**CDKN2A**	Loss of function/ Deletion	58%		[9]
**CASP8**	Inactivation	11%	Apoptosis evasion	[9]
**PIK3CA**	Gain of function/Amplifications	34%	Increased proliferation and pro-survival signals	[9]
**MYC**		14%		[9]
**CCND1**		31%		[9]
**Tyrosine kinase receptors**		60%		[9]
**TP63**	Copy number gain	19%	Pro-survival signals and cell differentiation inhibition	[9,11,12,13]
**SOX2**		n.d.		[11,12,13]
**NOTCH pathway**	Inhibition	26%		[15]
**FAT1**	Loss of function	25–30%	Promotion of tumor progression and cancer stemness	[16]
**NRF2/KEAP1 pathway**	Activation	20%	Increased celular detoxification	[17]

### 1.2. Available Therapies for HNSCCs

In the early stages (I and II), the treatment of HNSCC patients is mostly surgical. It can be accompanied by radiotherapy (RT), or concurrent cisplatin-based chemotherapy for patients that have locoregionally advanced tumors with closed or positive margins, among other factors that increase the risk of local recurrence (Table 2). These include perineural invasion, lymphovascular invasion, and extranodal extension [2,18]. The detection of multiple affected lymph nodes is likewise an indication for RT.

Advanced disease (stages III and IV) will require surgery and multimodal treatments that vary between RT plus chemotherapy [19], pre-surgical RT or chemotherapy, and a combination of RT with the epidermal growth factor receptor (EGFR) monoclonal antibody cetuximab [20]. The main chemotherapy given is cisplatin or its analog, carboplatin, but 5 fluorouracil (5-FU) is also indicated. In these stages, chemotherapy also can be combined with paclitaxel. 

Recurrent, unresectable or metastatic diseases with no RT or surgical options will be treated with first-line monotherapy (cisplatin/carboplatin, docetaxel, paclitaxel, 5-FU, Cetuximab or gemcitabine), combinations of cisplatin/carboplatin with any of the monotherapies mentioned above, or combinations of cisplatin/carboplatin plus cetuximab with another chemotherapeutic agent of choice [21,22,23,24,25,26,27]. However, these treatments have limited beneficial effects and are usually accompanied by high toxicities. 

As an alternative first-line option, the immune checkpoint inhibitors (anti-PD-1) nivolumab and pembrolizumab have been approved for unresectable or metastatic disease. In a pre-specified exploratory analysis [28,29,30,31], OS was increased in nivolumab-treated patients with a PD-L1 combined positive score (CPS) ≥1 (8.7 versus 4.6 months,). However, OS was not significantly increased in patients with a PD-L1 CPS <1. Although treatment response to nivolumab and pembrolizumab are lower than initially expected (around 30%) and comparable to cisplatin-based combinatorial treatments, these results are still promising because around 10% of the patients developed a long-lasting response. 

Other alternative treatments are under investigation due to the limited benefits of currently available therapies. These therapies are based on the identification of the tumor drivers in a patient-specific manner, mimicking what is used in other more studied cancers. In patients with HNSCCs containing high frequencies of mutant HRAS variant alleles, the farnesyltransferase inhibitor tipifarnib demonstrated objective response rates of up to 56% in phase II trials [32]. These data are encouraging, although further studies are necessary before incorporating this approach into routine clinical practice. Furthermore, a number of small molecule tyrosine kinase inhibitors have been tested in patients with advanced HNSCC, including EGFR inhibitors, afatinib [33], gefitinib [34], erlotinib [35], and lapatinib, VEGFR and PDGFR inhibitor sunitinib [36,37], or the MET inhibitor tivantinib [38]. However, no clear clinical role has yet been established for this approach. The clinical role of palbociclib, a cyclin-dependent kinase (CDK) inhibitor specific to CDK4 and CDK6, remains investigational in patients with recurrent or metastatic HNSCC. Palbociclib has been evaluated in combination with cetuximab in patients with HPV-negative disease resistant to platinum and/or cetuximab [39]. While initial trials suggest that palbociclib plus cetuximab could potentially reverse previous cetuximab resistance, a randomized phase II trial (PALATINUS) demonstrated that this combination did not improve OS in patients with platinum-resistant disease [40].

Since cisplatin has been, and still is, the standard choice of treatment for advanced HNSCCs, we will focus this review on describing the main mechanisms of resistance to cisplatin, including both intrinsic mechanisms and adaptative tumoral mechanisms, as well as the composition of the tumor microenvironment (TME) and how it affects the response to chemotherapy.

### 1.3. Cisplatin Mechanisms of Action

Cisplatin is a platinum-based anticancer agent which, together with its derivatives, carboplatin and oxaliplatin, has been largely used to treat HNSCC patients and other solid tumors with significant efficacy. However, cisplatin treatment also causes toxicity-related symptoms (nausea, hair loss, and nephrotoxicity, among others [41]), while tumors often develop mechanisms of resistance, limiting its therapeutic benefits. In mammals, cisplatin uptake is mediated by the copper membrane transporters 1 and 2 (CTR1 and CTR2) [42]. Once cisplatin reaches the cytoplasm, chlorine atoms are displaced by water molecules to give rise to the active form. This hydrolyzed product is a potent electrophile that can react with any nucleophile group, including the sulfhydryl groups in proteins (such as those within reduced glutathione and metallothionein) and nitrogen donor atoms in nucleic acids [43]. It tends to establish covalent bonds with the N7 reactive center in purine nitrogenous bases in DNA molecules, preferentially guanine bases [44]. Several types of DNA adduct can be generated from the reaction of cisplatin with DNA, including monoadducts, intra-strand and inter-strand crosslinks. If a low amount of DNA damage is caused, it can be completely repaired by the cell’s DNA repair mechanisms. However, if the amount of DNA damage accumulated exceeds the DNA repair capacity, cell death through apoptosis will be triggered. In addition, cisplatin increases reactive oxygen species (ROS) production, causing strong cellular stress that strengthens the apoptotic pathways mediated mainly by pro-apoptotic proteins such as Bax or Bak [43]. The contribution of other cell death pathways such as ferroptosis will be further discussed.

Because the principal mechanism of action of cisplatin is the formation of DNA–cisplatin adducts, enhanced DNA damage repair or the evasion of cell death are key to developing resistance to cisplatin.

## 2. Classical Mechanisms of Cisplatin Resistance

### 2.1. Control of Cisplatin Import and Export

Drug import and export were among the first studied mechanisms of cisplatin resistance, since decreased uptake or retention were the most obvious ways by which tumor cells could keep cisplatin at subtherapeutic concentrations (Figure 1A). CTR1 and CTR2 were shown to participate in cisplatin import and export. Studies carried out in non-HNSCC models showed that CTR1 expression increased cisplatin accumulation by 2.2-fold, while unexpectedly, CTR2 deletion increased cisplatin content by 9.1-fold [45], through increased Rac1-cdc42-dependent micropinocytosis. This last result was clinically validated, since high CTR2 expression in ovarian tumors had a poor outcome after cisplatin treatment. In a study using the TCGA cohort of HNSCCs, the authors could not detect a significant correlation between decreased *CTR1* expression and residual tumors after first line chemotherapy. The presence of residual tumor cells was used as a read out of chemotherapy resistance. Instead, this study identified a decreased expression of the transporter *VRAC* (*LRRC8A*) [46] in tumors with residual disease after chemotherapy. Deletion of *VRAC* in a laryngeal SCC line drastically increased its resistance to cisplatin. In an independent study, the expression of *CTR1* or *ATP7B* transporters did not correlate with cisplatin IC50 values or with the accumulation of cisplatin and the presence of DNA adducts [47], supporting the previous results. Regarding CTR2, there are no studies describing its function in HNSCCs, but analysis of the TCGA shows that it is deleted in 4–5% of HPV-negative HNSCCs. In addition, upregulated expression of the transporter OCT3 (*SLC22A3*) increased cisplatin uptake and its cytotoxicity [48]. Finally, the function of other transporters such as ATP7A/B, MRPs or ABCs has been correlated with cisplatin resistance, but their specific function in HNSCCs remains to be studied [49].

### 2.2. Mechanisms of DNA Repair

Once cisplatin has entered the cell, it will cause DNA damage and oxidative stress. To maintain genetic integrity upon exposure to cisplatin, cells need to activate DNA repair pathways to remove platinum-DNA adducts (Figure 1B). This process is complex and tightly regulated in cells, involving complementary DNA repair mechanisms that guarantee the success of the process to prevent cell death. These mechanisms are hyper-activated in cancer and, for many years, have been known to guide resistance to chemotherapies such as cisplatin. In HNSCCs, DNA repair mechanisms and loss of genetic integrity are also coupled with tumor initiation and progression, and HNSCC tumors often carry mutations in at least one DNA repair gene. These errors in DNA repair are thought to drive tumor initiation by generating tumor driver events such as TP53 mutation, CDKN2A deletion or PIK3CA amplifications. For instance, Fanconi anemia (FA) patients that carry germline mutations in components of the DNA repair FANC pathway have a 500–700-fold increased risk of developing HNSCCs [50]. On the other side of the coin, once tumors are established, they can activate specific DNA repair pathways to deal with chemotherapeutic insults, but the repairs may not be precise and might lead to the accumulation of other mutations. One could hypothesize that these new mutations will also be involved in the development of cisplatin resistance [51,52], but more detailed studies need to be conducted to prove this hypothesis in the context of HNSCCs. 

The nucleotide excision repair pathway (NER) [53] is responsible for repairing intra-strand platinum-DNA adducts, which is the main type of DNA damage caused by cisplatin. This process is guided by a complex machinery that includes around 30 proteins required for activities from the recognition of DNA damage to base excision and filling of the gap. Among these NER-associated proteins, DNA excision repair proteins ERCC-1 and ERCC-4 (also known as XPF) have a critical role in inducing cisplatin resistance [54]. These two enzymes are endonucleases that form a complex to catalyze the removal of platinum adducts, which is the rate-limiting step of the NER process. Two studies showed that increased expression of ERCC1 [55] and XPF [56] can predict clinical response to cisplatin in HNSCCs. Higher ERCC1 expression was detected in 26 out of 57 patients, and high ERCC1-expressing tumors produce a lower patient progression-free survival and OS rate [55]. XPF was detected at higher levels in the oral cavity, compared with the oropharyngeal or larynx regions [56], where higher expression of XPF correlated with lower progression-free survival. Both studies suggested that an increase in the NER pathway and DNA repair capacity can produce resistance to cisplatin and predict poor prognosis in patients with HNSCCs.

When the NER pathway fails to repair the adducts, potentially lethal double-strand breaks will be produced. The homologous recombination pathway (HR) is the main system responsible for repairing DNA lesions [57]. This system is composed of BRCA1/2 proteins (FANCS and FANCD1, respectively), whose mutations have been associated with breast and ovarian familial cancer development. However, these patients do not have a predisposition to develop HNSCCs. On the other hand, mutations in FANC genes that correct inter-strand crosslinks caused by cisplatin are the main cause of FA, and as we mentioned before, patients with this condition have a greater risk of developing HNSCCs. The implications of defects in FA/HR pathways in cisplatin response were examined in a comprehensive cohort of 29 patient-derived HNSCC cell lines. It was shown that some cell lines derived from sporadic HNSCCs behave similarly to FA-HNSCC-derived cell lines, and have hypersensitivity to DNA crosslinking agents and functional crosslinking repair defects. Analysis of patient samples detected rare germline and somatic variants in FA/HR pathway genes in 23% of the analyzed cell lines and in 19% of the tumors studied. Patients with these defects had a lower OS. Tumors harboring FA/HR variants demonstrated a clinical benefit with high cumulative cisplatin doses, but lower OS if the dose of cisplatin given was low. The variants in question were found in the BRCA1/2, FANCA, FANCG and FANCD2 genes. Another study showed that mutations in HR genes (*ATM*, *BRCA1/2*, etc.) were present in 17.6% of a cohort of 170 patients with HNSCCs. These mutations were more frequent in laryngeal cancer, older patients or patients that had received radiotherapy and/or chemotherapy previously, presenting poor outcomes. Importantly, that 17.6% of patients had mutations considered “deleterious” or “inactivating”, which made these patients candidates for treatment with PARP1/2 inhibitors [58]. 

Additionally, the O6-methylguanine DNA methyltransferase (MGMT), which repairs alkyl adducts at the O6-position of guanine, is a well-known regulator of cellular resistance to O6-alkylguanine alkylating agents such as temozolomide [59]. MGMT-proficient nasopharyngeal SCCs are more resistant to cisplatin than MGMT-deficient tumors, since MGMT can also bind to platinum-DNA adducts, reinforcing DNA repair after cisplatin treatment. In fact, high levels of MGMT can also predict decreased patient survival [60]. Development of MGMT inhibitors raises the possibility of applying them in combination with cisplatin to improve patient response. Due to the involvement of the different DNA repair enzymes in the development of cisplatin resistance, their use as cisplatin-response biomarkers has been proposed repeatedly. Their use has not been extended due to the lack of alternative combinatorial treatments that could improve cisplatin response in the context of DNA damage gene mutations; this classification has no impact on treatment choice, and patients will still receive cisplatin treatment regardless.

### 2.3. Cellular Detoxification of Reactive Oxygen Species

For many years, the predominant cause of cell death from cisplatin treatment was thought to be from the accumulation of DNA damage. However, recent studies have highlighted the importance of the accumulation of oxidative stress, particularly ROS and the mechanisms of detoxification in developing cisplatin resistance.

Cisplatin-induced oxidative stress is caused by the mitochondrial release of ROS into the cytoplasm (Figure 1C). Accumulation of high ROS levels may damage lipids, proteins and DNA to an extent that can induce cell death. To deal with ROS, cells have a complex scavenging system composed of superoxide dismutase (SODs), glutathione peroxidase (GPX), glutathione reductase (GR), peroxiredoxin, thioredoxin and catalase. In response to an increase in ROS, the expression of these genes is rapidly induced by the TF nuclear factor erythroid 2-related factor 2 (*NFE2L2*, henceforth referred to as *NRF2*), considered a master regulator of cell detoxification. Increased NRF2 expression is a common event in cancer progression, which can be caused by constitutive transcriptional activation or a gain in gene copy number, but the increase in its protein level is usually caused by silencing, deletion, or mutation of its inhibitor Kelch-like ECH-associated protein 1 (KEAP1). NRF2 expression is increased in HNSCCs, where it drives malignant growth not just by controlling ROS levels, but also by regulating nucleotide biosynthesis [61]. 

In the context of chemotherapy resistance, cisplatin-resistant HNSCC cells express higher levels of NRF2 protein [62,63]. NRF2 is directly involved in promoting cisplatin resistance by maintaining a high expression of ROS scavenging proteins, which prevents the accumulation of ROS in steady state conditions, preventing cancer cells from dying [63]. Mechanistically, NRF2 expression has been shown to be dependent on EpCAM-IL6/p62 expression [63] and also on c-MYC gene expression [61], which are both increased in HNSCCs. Several compounds, such as wogonin, have been used to inhibit NRF2 expression in cisplatin-resistant cells, reducing NRF2 protein and reduced-glutathione (GSH) levels and promoting cell apoptosis of cancer cells without affecting healthy ones [62].

An alternative cell death pathway caused by NRF2 inhibition is ferroptosis. Ferroptosis is programmed necrosis mainly triggered by extra-mitochondrial lipid peroxidation arising from iron-dependent ROS accretion [64]. Combined cisplatin treatment with the induction of ferroptosis by the chemical inhibition of cysteine/glutamate antiporter SLC7A11 using sulfasalazine [65] re-sensitized HNSCC cell lines to cisplatin, by decreasing reduced GSH. This synergistic effect was shown in vitro, and in vivo in preclinical mouse models, where combined treatment with cisplatin and sulfasalazine produced a significant reduction in tumor growth. Furthermore, inhibition of glutaredoxin 5 (GLRX5), which is responsible for transferring free iron to iron-containing proteins, upregulated the iron starvation response, increased intracellular iron levels, and synergized with ferroptosis induction in cisplatin-resistant HNSCC cells [66]. The induction of ferroptosis to avoid NRF2-driven cisplatin resistance has appeared as a promising therapy due to the specific upregulation and seeming addiction of NRF2 in cancer cells, and the development of NRF2-specific inhibitors or downstream pathways could open new possibilities for the treatment of HNSCC patients.

## 3. Tumor Heterogeneity: Cancer Stem Cells and Therapy Resistance

One of the major mechanisms of the development of therapy resistance is the presence of phenotypically distinct subpopulations of cancers cells, which is known as intra-tumor heterogeneity, henceforth referred to as tumor heterogeneity (TH) [67]. TH can be driven by the presence of clones carrying different mutational events, which could respond differently to treatments, or by cells containing the same mutations, which are phenotypically distinct due to metabolic or epigenetic reprograming. Within the latter, cancer stem cells (CSCs) have been shown to be key drivers of therapy resistance by surviving treatments and driving tumor relapses (Figure 2). It has been demonstrated that CSCs can originate from the accumulation of mutations in tissue specific stem cells (SCs) that already possess increased self-renewal capacity [68]. 

HNSCCs originate from hierarchically organized epithelia, which are maintained by adult stem cells. Although there is no strong evidence that all HNSCCs originate from tissue adult SCs, due to its similarities with the skin epithelium (where the cells of origin of cutaneous SCCs are well defined as K14-expressing stem cells [69]), it is likely that HNSCCs share the same origin following the cancer stem cell hypothesis [70]. However, adult SCs of the different mucosa in the head and neck region have not been identified properly, except for the tongue mucosa [71]. 

In the mouse tongue epithelium, progenitor cells are organized into a basal layer, in contact with the underlying muscle or connective tissue, and can proliferate to self-renew, due to the expression of the TFs SOX2, PITX1 and TP63 [72], or to terminally differentiate (Figure 2A and Figure 3A). Among the first two to three layers of basal cells of the tongue epithelium, there is a population of slow-cycling cells that express B cell-specific Moloney murine leukemia virus insertion site 1 (BMI1). BMI1 is an E3-ubiquiting ligase that mono-ubiquitinates histone H2A and functions as a catalytic component of the polycomb repressive complex 1 (PRC1). BMI1 controls self-renewal in several adult SCs, where it acts by repressing the expression of cell cycle inhibitors and driving proliferation and survival [73]. However, its expression is also increased in many cancer types, where it drives the self-renewal properties of cancer cells, by repressing the expression of tumor suppressor genes [74]. 

In the oral epithelium, BMI1-positive SCs are activated upon radiation-induced damage to proliferate and repopulate the injured tissue, suggesting that they behave as tissue stem cells [75]. Lineage tracing analysis upon the induction of chemical carcinogenesis with 4NQO on oral epithelial cells and development of SCCs identified some tumor cells also expressed BMI1 (Figure 2C) [76]. In those tumors, BMI1-positive cells are slow-cycling cells with increased self-renewal capacity and have shown increased capacity to form invasive SCCs that metastasize to lymph nodes [77]. Moreover, BMI1-positive cells persist upon treatment with cisplatin, driving SCC recurrences. However, if BMI1-positive cells are targeted with diphtheria toxin or using a BMI1 inhibitor (PTC-209), cisplatin treatment eliminates oral SCC (OSCC) lesions and prevents the formation of lymph node metastasis. This confirms that BMI1-positive cells could act as CSCs and could additionally cause the appearance of cisplatin refractory tumors. Mechanistically, BMI1-positive cells express classical CSC signatures and adhesion molecules which explains their aggressiveness. They also express the signature of stress and inflammation markers, driven by AP1 TFs, which are also critical for driving persistence upon cisplatin treatment (Figure 3B). 

In human HNSCC patient samples, the increased expression of BMI1 correlated with a decreased recurrence-free probability. BMI1 expression can be increased by IL6 secreted by endothelial cells [78] and by cisplatin treatment through the induction of IL6 expression by the cancer cells themselves (Figure 2B) [79]. This suggests that HNSCCs may respond to cisplatin by increasing the pool of CSCs, which could partly explain the existence of residual disease. The use of the IL6R monoclonal antibody tocilizumab prevented BMI1 upregulation by blocking IL6 function, demonstrating a reduction in tumor growth in subcutaneous injected xenografts [78]. 

Overall, these data are promising, and point to BMI1 as an appealing target to treat advanced or cisplatin refractory HNSCCs. Still, how BMI1-positive cells are regulated and the existence of resistance mechanisms to BMI1 inhibition is unknown. Other well-known markers of CSCs, such as CD133 [80,81], CD44 [82], EpCAM [63], and ALDH1 [83,84], are expressed by small populations of HNSCCs cells (Figure 2C) [85]. In most cases, these studies have been carried out in cell lines, to show that these markers are critical for sphere formation, cisplatin resistance, and tumor initiation in immunodeficient mice. Although their clinical relevance as prognostic markers has also been described, their function in a more physiological context remains to be addressed.

Some key properties of CSCs include metabolic changes compared to non-CSCs, higher expression levels of NRF2-dependent detoxifying machinery, expressing mesenchymal programs or the possibility of existing in a slow cycling, quiescent state (Figure 2C). Importantly, all these characteristics are frequently present simultaneously, making CSCs very efficient at escaping chemotherapy and regrowing tumors after treatment [86,87]. In line with these characteristics, Ching-Wen et al. showed that HNSCCs have heterogeneous ROS content, with cells having high or low ROS levels [88]. ROS-Low HNSCC cells are less proliferative compared with ROS-High ones, and have higher tumor initiating capacity when transplanted in immunodeficient mice, suggesting that ROS-Low HNSCCS have CSC characteristics. In accordance with this, ROS-Low CSCs express higher levels of antioxidant genes SOD2 and CAT and as mentioned above were much more resistant to chemotherapy. 

Additionally, we identified that cutaneous SCCs (cSCCs) xenografts possessed proliferative heterogeneity and contained a population of quiescent cells [89]. Quiescence is a cell cycle state of prolonged but reversible cell-cycle arrest. Quiescence has emerged as a mechanism to resist chemotherapy [90], since chemotherapy targets rapidly proliferating cells by creating DNA damage. Quiescent cells are thought to either increase DNA-damage repair capacity or their cell cycle arrest properties give them a longer time to repair their DNA, preserving their genomic integrity. In cSCCs, quiescent cells were also more efficient at forming tumors in limited dilution experiments. The quiescent phenotype was under control of the TGFβ/SMAD pathway. We demonstrated that quiescent cells are responsible for tumor residual disease upon chemotherapy with 5FU in cSCCs. Finally, we showed that TGFβ inhibition in HNSCC cultured cells and can increase sensitivity to cisplatin treatment, and that patient HNSCC tumors that progressed after treatment were enriched in a transcriptional signature of cell quiescence. In cSCCs, all cells with tumor propagating capacity already had an elevated NRF2 signature, but it is possible that this is not the case with HNSCCs, where low ROS may occur only in quiescent cells or other CSC populations. Our data suggested that HNSCCs contain a population of quiescent cells maintained by TGFβ, but more detailed experiments will be needed to clarify their role in mediating therapy resistance in an in vivo setting.

Eliminating CSCs to improve therapy response is not a novel idea, and the scientific community has been exploring these options for decades, but it has appeared to be quite challenging [91]. The success of these approaches has been limited by both TH itself as well as the presence of different pools of CSCs that can adapt to the treatment due to their plasticity, which will be discussed in the next section. Regardless, promising targets such as TGFβ and NRF2 pathways or inhibitors of other molecules such as ALDH1 [84] are being tested in investigational or clinical trials and could be combined with cisplatin-based chemotherapy to improve patient outcome. 

## 4. Epithelial to Mesenchymal Transition and Cell Plasticity

One of the key events that promotes tumor progression is the acquisition of migratory capacity by tumor epithelial cells, allowing them to invade, extravasate and, in the end, form metastases. This requires a process known as epithelial-to-mesenchymal transition (EMT). EMT is considered as a cellular reprograming event by which epithelial transcriptional programs are inhibited and mesenchymal ones are activated. Once the cells arrive at the metastatic site, they need to revert to an epithelial state in a process named mesenchymal to epithelial transition (MET) to be able to grow and form metastases [76]. Although these processes have been identified and well-described in vitro, the identification of cancer cells with full EMT characteristics in patient specimens has failed [92]. Novel studies recently conducted in vivo have demonstrated that cancer cells do not experience a complete EMT; instead, EMT seems to be a partial and flexible reprograming event [92]. This partial or hybrid EMT (pEMT) induction represents a cell state where cancer cells will express mesenchymal (such as vimentin) and epithelial (such as E-cadherin) genes at the same time. Due to epigenetic plasticity, epithelial gene expression can decrease just enough for the cells to lose adherence and migrate, while retaining the necessary epithelial characteristics to re-grow as an epithelial tumor at the metastatic site in an efficient manner. This phenotype and its plasticity are regulated by the activity of TFs mediating interactions with DNA and changing patterns of gene expression, as we will explain in detail later. Hybrid EMT cells will express epithelial transcription factors such as *TP63* or *SOX2*, and mesenchymal ones, such as *SNAIL/SLUG*, *ZEB1*/2 or *TWIST1*/2, in a way where the expression of genes required for migration or growth can switch on demand depending on the cancer cell’s needs [93]. One of the key inducers of EMT is TGFβ, that can be secreted by cancer cells or by different populations of the TME, as we will discuss in detail.

Interestingly, the EMT transcriptional program has consistently been linked to the CSC phenotype and to drug resistance (Figure 2C) [67,94]. CSCs and drug-resistant CSCs will frequently express high levels of EMT signature genes. For instance, BMI1-positive cells express lower levels of epithelial differentiation genes and higher migratory genes such as some metalloproteases (MMPs), facilitating the remodeling of the extracellular matrix (ECM) and the invasion of the adjacent tissue [77]. Although the link between the induction of EMT (by overexpression of EMT TFs) and the acquisition of CSC properties is well-established, the mechanisms behind the increase in stemness are not well understood [67,94].

Puram et al. characterized the tumor heterogeneity and transcriptional programs of tumor cells from primary and lymph node metastasis from HNSCCs at the single cell level, uncovering a population of cells expressing a transcriptional program of pEMT [95]. The presence of this population of cells predicted nodal metastasis and adverse pathological features. Much like complete EMT, pEMT expression and invasive capacity of pEMT expressing cells is under the control of TGFβ signaling. Interestingly, not all classical EMT TFs correlated with the pEMT signature—*SNAI2* (also known as SLUG) did, but *SNAI1* (SNAIL), *ZEB1*/2 and *TWIST1*/2 did not. This was the first time that partial EMT expressing cells were described in HNSCCs, and although the connection with the metastatic process was shown, the effects of the pEMT phenotype on CSC properties were not studied. The fact that pEMT cells were detected at the basal layer of the tumor, where adult SCs are usually localized, may suggest that pEMT could indeed be enriched in CSC properties, but further analysis would need to be conducted to prove it. 

Furthermore, it has been shown that EMT induction modulates some of the mechanisms related to drug resistance by increasing the expression of genes that induce drug efflux, decreasing cell proliferation, or blocking pro-apoptotic pathways [67]. EMT-expressing cells also have increased expression of DNA repair mechanisms to repair DNA damage and survive. In fact, the EMT TFs SNAI1, SNAI2 and ZEB1 have been shown to be involved in mediating DNA repair in response to DNA damage [96]. SNAIL controls *ERCC1* expression, and together they mediate cisplatin resistance [97]. ZEB1 functions in DNA repair downstream of ATM [98]. SLUG has a function in the HR pathway, where also it participates downstream of ATM, indicating DNA damage and ensuring proper repair. It has been shown that SLUG is required for proper DNA damage resolution, and its deficiency led to the accumulation of mutations and tissue aging [99]. In OSCC cell lines, SLUG interference increased the sensitivity of OSCCs RT [100], although its role in mediating cisplatin resistance has not been evaluated in detail. Since SLUG has been proposed as a key regulator of pEMT [101], it is likely that it could mediate chemotherapy resistance in advanced HNSCCs at least by increasing DNA repair after treatment. On the other hand, although ZEB1 and SNAIL expression does not correlate with pEMT, their function in DNA repair suggests it could have a more general function unrelated to the pEMT phenotype. 

There is overwhelming evidence of the link between EMT and chemotherapy resistance, but in epithelial tumors, EMT and chemoresistance can be separate processes. TH and how the cells respond transcriptionally to a chemotherapeutic insult will determine the obtained resistant phenotype [102]. Sharma et al. showed that tumor epithelial cells can adapt to cisplatin in two ways depending on the degree of TH. Homogenous epithelial tumors can undergo a reprograming event, changing their phenotype to mesenchymal. Heterogenous tumors composed of epithelial and mesenchymal-like cells will become positively selected for the cell phenotype that best overcomes resistance. Resistance mechanisms in these two situations will rely on different stemness-related TFs, and while resistant epithelial tumors rely on the retention of SOX2 activity, mesenchymal tumors will be dependent on SOX9 [102]. They demonstrated that the reprograming into a mesenchymal phenotype depends on BRD4 activity and the ability of tumor cells to rewrite their enhancer landscape to adapt to the treatment. In fact, fully epithelial tumors that triggered a mesenchymal phenotype contained poised promoters at mesenchymal genes, such as vimentin and IL6, becoming fully activated by H3K27Ac marks deposited by the BRD4 complexes in response to cisplatin treatment (Figure 3C). 

These data reflect the complexity of the adaptation mechanism to therapy, complicating the use of epithelial and mesenchymal markers to predict chemotherapy response, and may even explain why the efforts made to translate this knowledge to the clinic have failed. A different matter is the use of these markers to predict the appearance of local or distal metastasis, since the connection between the expression of EMT markers and the presence of metastasis at the clinical level has been well-described.

## 5. Epigenetic Mechanisms of Cisplatin Resistance

Epigenetics are defined as the mechanisms that control gene expression without modifying the sequence of the DNA. Early studies in cancer biology defined DNA mutational events as unique tumor drivers, but since the discovery that these mutations are already present at clonal levels in healthy tissues [103], it has become widely accepted that other non-genetic alterations are needed for premalignant lesions to arise. Lifestyle choices such as smoking, drinking, or other insults to the upper respiratory epithelia can cause inflammation, which has been shown to alter epithelial cells in other tissues, and can leave epigenetic scars that may affect future cellular responses and prompt future tumor initiation [104]. In addition, copy number gains or mutations in epigenetic regulators can alter the regulation of gene expression and promote the acquisition of malignant transcriptional programs that can also allow adaptation to therapies, as we just described with the EMT phenotype. Here, we will focus on epigenetic or non-genetic mechanisms affecting DNA/histone modifications as well as TFs, and their role in tumor progression and development of therapy response. 

Among the genes altered in HNSCCs are some chromatin modifiers such as *KMT2D*, *NSD1*, and *ACTL6A* and TFs such as *TP53*, *NFE2L2* (*NRF2*), *MYC*, *TP63* [11,12] and *SOX2* [105], among many others. Although there are not many studies in HNSCCs, we will summarize the potential mechanisms of tumor progression based on the similarities to other cancers. Mutations in TP53 that inhibit its pro-apoptotic functions occur in HNSCCs, but these mutants can still activate other targets such as chromatin modifiers *MLL1* (*KMT2A*) and *MLL2* (*KMT2D*), altering enhancer activities [106]. The absence of wild-type TP53 also has metabolic consequences that decrease 5hmC DNA methylation marks that can again affect enhancer and promoter activities [107]. These changes in enhancer activity or the activation/repression of specific enhancers will be a key event in deciding the cell fate of premalignant cells. Mutation or deletion of the histone methyltransferase *KMT2D*, which deposits the active H3K4me1 mark [108], is present in 17% of HNSCCs, and leads to specific loss of enhancer activity in IRF7/9-driven enhancers (Figure 3B). IRF7/9 TFs control the expression of cytokines that impact numerous immune-signaling pathways, such as interferon signaling, critical to producing effector T cells. Therefore, mutations in *KMT2D* can create an immunosuppressed TME, as seen in late-stage HNSCCs. Mutations of *KMT2D* could endorse tumor growth by promoting immune system escape, contributing to a poor response to immune and chemotherapies. Nuclear receptor binding SET domain protein 1 (*NSD1*) is a histone methyltransferase altered in 10% of HNSCC HPV-negative patients. Patients with NSD1 alterations have a concomitant loss in H3K36me2 and global DNA hypomethylation in intergenic regions, but also a decrease in H3K27Ac active marks at regulatory elements, which leads to a decreased expression of their controlled genes [109]. These genes affect important pathways such as KRAS, EMT or inflammation, all of which are involved in chemotherapy response, and therefore NSD1 mutant cell lines are more sensitive to cisplatin treatment [110]. Finally, BMI1 itself is a component of the PRC1-repressing complex, and acts together with the PRC2 complex to repress the promoter of tumor suppressor-related genes to enhance the CSCs phenotype (Figure 2 and Figure 3B). 

Although chromatin modifiers are critical in controlling DNA accessibility and enhancer activity, TFs can act as pioneer factors, guiding and promoting chromatin opening and are essential to promote enhancer/promoter interactions to drive gene expression (Figure 3B). Tumors usually acquire the expression of cancer-specific TFs [14] that are not expressed in healthy tissue, and at the same time hijack tissue-specific TF functions to ensure their survival. In this way, newly expressed TFs can activate alternative enhancers, or use already accessible ones to promote tumorigenic transcriptional programs (Figure 3B). For example, SOX11 [111] and SOX8 [112] TFs are not detected in the oral epithelium, but they become overexpressed in recurrent and cisplatin-non-responsive HNSCCs. Overexpression of both TFs drives resistance to cisplatin by enhancing EMT characteristics. On the other hand, *TP63* [13] and *SOX2* [105] amplifications boost self-renewal programs, as they do in the skin and healthy oral epitheliums (Figure 3A). Amplifications in the chromatin modifier *ACTL6A* occur in 20% of HNSCCs, co-occurring with *TP63* amplifications, and its overexpression has been linked to chemotherapy resistance. ACTL6A/TP63 drive the activity of YAP by repressing YAP inhibitor WWC1, which, as we will explore in detail in the following sections, results in poor prognosis [12]. 

Additionally, chromatin modifiers can have transcription-independent functions. ACTL6A cooperates with the SWI/SNF protein complex and its chromatin remodeling activity to repair cisplatin DNA adducts. More importantly, ACTL6A-driven cisplatin resistance could be reverted using HDAC inhibitors. Finally, as we previously mentioned, enhancer plasticity driven by BRD4 activity is key for HNSCCs adapting to epithelial or mesenchymal phenotypes and persists after cisplatin treatment (Figure 3C) [102]. 

Although much more detailed studies are needed, these publications exemplify the relevance of epigenetics in cisplatin resistance. Other mechanisms such as miRNA activity, or DNA methylation, are also important regulating therapy-resistance mechanisms. For example, MGMT promoter methylation can functionally predict temozolomide response in glioblastoma [113], while miRNA-7 promoter methylation status can predict response to cisplatin in ovarian cancer [114]. 

## 6. The Tumor Microenvironment and Therapy Resistance

Since the establishment of the first cell line in 1951, scientists have used 2D culture models to study the intrinsic properties of cancer cells, trying to unravel mechanisms of chemotherapy resistance. Nevertheless, tumor cells grow in a continuously changing ecosystem within 3D structures that contain nutrient and oxygen gradients, and constantly interact with biological factors such as other cell types in a physical space formed by the ECM. All these factors will change the cancer cell characteristics and add other players to the equation of chemotherapy resistance. These elements can be classified into physical and biological factors (Figure 4). Physical components that interfere with the delivery of cisplatin include high cell density, fluidic shear stress, and the ECM. The biological components are the consequences of cell tumor growth (hypoxia or acidity) and non-tumoral cells (cancer-associated fibroblasts, macrophages, or T cells).

### 6.1. Physical Factors: ECM Remodelling, Composition and Stiffness

A growing body of evidence suggests that the ECM supporting the growing mass of tumoral cells can modulate therapeutic response. Research has shown that biophysical traits such as matrix density, stiffness and composition can alter drug delivery, reception, and efficacy. In fact, high ECM rigidity is often a poor prognostic factor and is associated with chemoresistance and higher metastatic potential in highly fibrotic tumors [115]. ECM stiffness and density are determined by its molecular composition, consisting of fluctuating quantities of fibrillar proteins such as collagens, fibronectin, laminins and elastins, which serve as ligands for cell adhesion molecules and provide structural scaffolding for cellular components in the TME. These molecules are deposited both by tumoral and stromal cell populations, which are involved in a complex interplay of secretory programs. In this context, enzymes such as MMPs, lysyl oxidases (LOX) and LOX-like proteins are deposited, contributing to ECM protein crosslinking, remodeling, and ultimately stiffening [116] (Figure 4A). HNSCCs have gained notoriety for having considerably rigid ECMs and high stromal infiltration; OSCCs in particular have been widely documented to have stiffened margins [117]. A tumor-induced gain in ECM stiffness has been observed in a wide variety of carcinomas, and several ways in which this can occur have been proposed. A shift in ECM protein composition is perhaps the beginning of a structural rearrangement that ultimately leads to a positive feedback loop among TME populations, thus generating and sustaining a protumorigenic environment. Cancer cells can recruit and “cancerize” ECM-embedded fibroblasts, which are primarily responsible for ECM protein deposition. These stromal cells can become activated by proximal tumoral cells in a bidirectional signaling exchange and reprogram their secretory and contractile functions to promote a tumorigenic ECM. These cells, known as cancer-associated fibroblasts (CAFs), will be discussed in further detail.

Upregulation and increased secretion of certain collagens by CAFs (particularly COL8A1 and COL11A1) seems to promote resistance to common chemotherapeutic agents such as cisplatin in HNSCC cell lines through collagen discoidin-domain receptor 1 (DDR1) upregulation, which controls cell differentiation and tissue homeostasis. Although its role in cancer is yet to be fully elucidated, growing evidence links this tyrosine-receptor kinase to metastatic and pro-survival mechanisms [118]. Other collagen receptors, such as integrins, which also bind ECM component fibronectin are overexpressed in the majority of cancers, including HNSCCs. Certain integrin subunits play a pivotal role by mediating cell-ECM signaling through focal adhesion protein (FAP) complexes. As matrices stiffen, focal adhesion complexes assemble in clusters, integrating and transducing biomechanical cues from the ECM, rearranging the actin cytoskeleton, and triggering Src, FAK, and PI3K/AKT downstream signaling pathways, among others [119]. Integrin β1 is especially noteworthy in the HNSCC context because it can bind a wide variety of ECM components and is commonly upregulated in metastatic HNSCCs as well as in other carcinomas [120,121], correlating with CSC-like phenotype acquisition and lower survival rates (Figure 4B). Heterodimeric complexes, such as integrin α4/β1 dimer, have been linked to cell adhesion-mediated drug resistance (CAM-DR) to EGFR inhibitors (cetuximab) [122]. Overall, the integrin-mediated interaction with the ECM triggers stemness-related mechanisms of cell survival.

Higher cellular density from increased tumor proliferation creates a progressively hypoxic environment due to a shortage of oxygen and available nutrients. Hypoxia inducible factor (HIF1α) target genes feed the stiffening ECM with the induction of LOX, LOXL, collagens, and MMPs that are responsible for basal membrane protein cleavage [123], releasing embedded growth factors (GFs) such as vascular endothelial growth factor (VEGF), promoting neovascularization [124,125]. This aggressive angiogenic program, activated to supply the growing tumor’s metabolic requirements, forms permeable vessels and inefficient intra-tumoral microcirculation, causing impaired drug delivery. Hypoxic transcriptional programs are thought to cause drug therapy resistance through the upregulation of ATP-dependent membrane transporter families such as ABC-type or MDR1, which reduce intracellular chemotherapeutic agent concentrations to subtherapeutic levels as we already mentioned [126]. Aberrant tumor vasculature and compression of lymphatic blood vessels increase interstitial fluid pressure and thus create higher shear stress [127]. This type of mechanical stimuli has been shown to promote tumor progression and acquisition of CSCs-like and EMT markers, as well as cisplatin chemoresistance by upregulating ABC-drug transporters such as ABCG2 and activating the PI3K/AKT pathway [128,129].

A stiff ECM has also been linked to the activation of EMT-related signaling pathways, which play a role in treatment resistance [130], as seen in pancreatic ductal adenocarcinoma (PDAC) and breast tumors. This promotes the translocation of transcriptional co-activator YAP1 to the nucleus, which is known to contribute to EMT program acquisition and has been detected to be overexpressed at the invasive front in HNSCC [131]. One of the hallmarks of EMT transition is the loss of cell–cell adhesion molecules (E-cadherin) and the upregulation of cell–ECM interactions (integrins, N-cadherin, FAPs), as well as canonical mesenchymal genes via the translocation of canonical EMT-triggering TFs (ZEB, TWIST, SNAIL) to the nucleus. A study found that OSCCs were mechanosensitive and reacted to collagen-rich rigid substrates by upregulating integrin signaling and focal adhesions, acquiring EMT-like expression profiles and thus correlating with more invasive phenotypes and lower recurrence-free survival [132]. This same study suggests that initially epithelial non-invasive OSCCs acquire EMT properties and become potentially metastatic after certain exposure to a stiff ECM, whereas more mesenchymal cell lines displayed innate invasive behavior. As cells became more invasive and acquired mesenchymal markers, it was shown that not only did cell–ECM adhesion proteins increase and stabilize, but they did so in an asymmetric manner, accumulating at the invasive front, showing greater FAK expression and localized, regularly deposited collagen. This correlated positively with an aggressive tumor phenotype and advanced disease stage in HNSCC cells cultivated in collagen matrices. In addition, mechanical cues, PI3K activation, certain GFs and integrin signaling can inhibit SNAIL degradation and permit EMT gene transcription.

Other ECM components such as small-size hyaluronan synthase-3 (HAS-3) have been linked to malignant processes in a variety of carcinomas, while isoform HAS-2 downregulation has shown cisplatin resensitization and tumor growth arrest in OSCCs. CSCs marker CD44 variant 3 (CD44v3) has been detected and linked to lymph node metastasis and cancer progression in HNSCC [133]. This alternatively spliced variant contains a hyaluronic acid (HA) binding domain and can also bind to other important GFs in the ECM (FGF, EGF), being involved in various tumorigenic processes. HA binding to CD44v3 has been proposed as an activating mechanism for target gene transcription of CSC markers Nanog, Oct4 and SOX2. SOX2 is also a direct downstream target of YAP/TAZ co-activators in the Hippo pathway, which, as mentioned before, are mechanosensitive and translocate to the nucleus when exposed to rigid ECMs [134].

### 6.2. Biological Factors

The cellular component of the tumor microenvironment can be divided into immune and non-immune cell types. The non-immune cell types are mainly fibroblasts and endothelial cells, and a minority of other important cell types of the neural lineage such as Schwann cells in tumor-associated innervation [135]. 

#### 6.2.1. Cancer-Associated Fibroblasts

Cancer-associated fibroblasts (CAFs) represent up to 70% of the stromal cells in the TME within HNSCCs [136], having both the ability to potentially promote or restrain tumor progression and metastasis [137]. These cells are highly molecularly heterogeneous and are still, despite current research efforts, vaguely characterized, which makes accurate identification and targeted-therapy development hard at best [138]. Although CAF subtypes and molecular signatures in HNSCCs are still ill-defined, there seems to be a consensus on classification in other cancer subtypes, such as PDAC or breast cancer, in which they differentiate between myofibroblast-like CAFs and pro-inflammatory CAFs which have a secretory phenotype [139,140,141]. The former displays a characteristic elongated morphology and contractile properties, upregulating the expression of certain markers associated with, but not exclusive to, this phenotypic profile, such as alpha smooth muscle actin (α-SMA, ACTA2 gene), FAP, FSP1 and PDGFR-a/b [141]. These are thought to intervene in ECM protein deposition and remodeling. On the other hand, inflammatory CAF functions include increased secretion of GFs and cytokines (IL-6, IL-11 and TGFβ) which recruit immunosuppressive cells to promote immune evasion [141]. The development of single-cell RNA sequencing technologies (scRNAseq) has recently allowed a better characterization of the CAF populations. In HNSCCs, this analysis revealed the existence of different populations of CAFs, including a myofibroblast population (defined by ACTA2 expression concomitant with IL6 expression) and a CAF population containing sub populations of TGFβ-related signature genes or ECM proteins [95]. Although the cell number of CAFs in this publication was limited, it showed that CAF phenotypes are more complex than the currently accepted phenotypes, and more detailed analysis will be needed to understand the specific role of each of these populations in tumor growth, chemotherapy, and immunotherapy responses in the HNSCC context.

TMEs enriched in CAFs are considered immunosuppressed and are also a predictor of bad prognosis [142]. In HNSCCs, CAFs’ presence surrounding cancer cells correlated with diminished patient OS. The effects of CAFs on chemotherapy response occur on various levels, including direct crosstalk and paracrine signaling with cancer cells, remodeling the ECM, and interacting with other cells from the TME to promote immunosuppression [143,144]. The most studied function of CAFs is the direct (cell–cell contact) or indirect (paracrine) induction of EMT in cancer cells [145,146,147]. Secretion of TGFβ or IL-6 by CAFs can also modulate the phenotype of CSC by triggering EMT or by inducing cellular quiescence, both being responsible for chemotherapy failure [148,149,150]. On the other hand, CAFs expressing BMP4 and low α-SMA in OSCCs support a higher proliferative tumor but contain a lower CSCs population, suggesting a higher sensitivity to chemotherapy [151]. In HNSCCs, scRNAseq uncovered that cancer cells located at the basal layer expressed a partial EMT, and they correlated these data with the presence of CAFs, but the relevance of their interaction on the induction of pEMT [95] and the response to chemotherapy was not investigated. A complementary study showed that CAFs producing higher levels of hyaluronan help to support tumor growth and tumor invasion. Additionally, another population of CAFs producing high levels of TGFβ were also critical in promoting tumor invasion, but did so with slightly less efficiency than hyaluronan-dependent CAFs [152]. Overall, these results demonstrate that different CAF phenotypes can support complementary tumor promoting mechanisms, while others can act by diminishing tumor aggressiveness. 

Chemotherapy administration and its effects are not tumor cell-exclusive, given that the agent penetrates within other proliferating cells in the TME. Compelling evidence strongly suggests CAFs respond to cisplatin, mainly through exosome, cytokine, and ECM component secretion, reducing therapeutic efficacy. Higher absorption and retention (and therefore decreased release) of cisplatin by CAFs have been linked to higher chemoresistance and recurrence of HNSCC, as well as greater clonogenic ability [153]. A rather understudied CAF-secreted cytokine, plasminogen activator inhibitor 1 (PAI-1), has been mentioned in previous studies as potentially being involved in cisplatin resistance after treatment, via apoptosis evasion. This cytokine activates AKT and ERK1/2 signaling pathways, inhibiting downstream caspase-3 mediated apoptosis and ROS accumulation [154]. Due to the relevant function of CAFs on both tumor progression and therapy resistance, this cell type is an attractive target to treat cancer. However, a much deeper understanding of its heterogenous function is needed to block exclusively pro-tumorigenic CAFs while leaving the suppressive ones unaffected. 

#### 6.2.2. Tumor-Associated Macrophages and Lymphocytes

Other understudied components of the TME in HNSCCs are the oral tissue-resident macrophages and tumor-associated macrophages (Figure 4C). Clinical trials are targeting these populations to optimize treatment because macrophages have proven to be strategic cells in the TME that can be hijacked by tumor cells to use in their favor to drive the disease. Macrophages communicate with T cells [155], fibroblasts [156] and cancer cells [157] while sending and receiving signals that are important in mediating cisplatin resistance. Tumor-associated macrophages (TAMs), just like CAFs, can promote angiogenesis, enhance stemness and EMT, remodel the ECM, and suppress the immune response (Figure 4C). Clinical data showed a positive correlation between an increased number of macrophages and poor OS [158], and specifically between the presence of CD163-positive macrophages and a poor response to chemoradiotherapy [159]. TAMs can interact with CAFs, which influences their own behavior to turn more immunosuppressive, correlating with a more invasive tumor type [160]. This causes the secretion of TGFβ and IL-6 among other factors, which will again result in the promotion of EMT and stemness in cancer cells. In fibrotic tumors such as pancreatic cancer and HNSCCs, CAFs will secrete numerous pro-inflammatory cytokines (CCL2, CCL3, CCL5, IL-6, GM-CSF, CSF-1, VEGF and CXCL8) that can affect recruitment, differentiation, and activation of TAMs [161,162]. Of particular relevance is the function of IL-6 that can also block macrophage differentiation into an immunosuppressive phenotype [163]. More detailed studies are needed to measure the cross-interactions between CAFs, TAMs and cancer cells specifically in HNSCCs to understand the potential vulnerabilities of this fatal combination that often composes a very aggressive tumor.

The presence of CAFs and TAMs usually correlates with the presence of regulatory T cells (Treg) and high expression levels of the inhibitory ligand PD-L1, which promotes T exhaustion [164]. Tregs can inhibit cytotoxic CD8+ T cells (responsible for the induction of tumor cell death) through the expression of TGFβ and other modulating factors such as IL-10 and IL-35 or through the expression of the inhibitory molecule CTLA-4. The presence of CD45RA^−^ Foxp3^high^ Tregs correlated with an advanced stage of HNSCC and a shorter relapse-free survival time after treatment with cisplatin or cetuximab [165]. The presence of low immune infiltrates also correlated with a poor response to cisplatin in breast cancer, although the specific mechanisms of action remain to be elucidated [166]. Overall, these tumor characteristics are considered immunosuppressive, correlating with EMT and stemness feature expression by cancer cells and higher invasive phenotype selection.

Cisplatin can also alter the immune landscape of the tumors, facilitating the appearance of tumor relapses. It has been shown that HNSCC recurrences contain decreased levels of B and T cell infiltrates and increased levels of macrophages, neutrophils, and dendritic cells [167]. This phenotype is more exacerbated in recurrences that were previously treated with chemoradiotherapy. A more detailed study demonstrated that cisplatin has both immune-enhancing and immunosuppressive effects, depending on the dose given. At low doses, cisplatin can enhance antigen presentation and T-cell cytotoxicity, but at high doses it may cause PD-L1 expression and impair T cell function [168]. This is extremely important, since the anti-PD1 antibody (nivolumab) is given as a first-line treatment to cisplatin-refractory tumors, and this may explain its low efficacy in HNSCC patients. To overcome these issues, some clinical trials are already testing the role of low doses of cisplatin as a neoadjuvant treatment in combination with anti-PD1 treatments.

## 7. Other Pathways Involved in Cisplatin Resistance

### 7.1. YAP/TAZ Pathway

YAP/TAZ proteins are downstream transcriptional coactivators within the Hippo pathway, which translocate from the cytoplasm to the nucleus and bind to DNA-binding TEAD factors [169]. Transcriptional regulation by Hippo pathway effectors such as YAP/TAZ proteins normally contribute to embryonic development, organ size, cell fate and polarity, stress, survival and tissue repair. These functions are normally dispensable in many adult tissues, and therefore YAP/TAZ proteins are typically constrained to the cytoplasm during normal homeostasis. Deregulation of YAP activity, however, has been robustly described in a wide array of human malignancies [170], linking its upregulation to oncogenic development and progression.

YAP/TAZ regulation is highly complex, integrating and responding to both cell-intrinsic and cell-extrinsic stimuli. Canonical Hippo pathway activation downregulates YAP/TAZ via LATS1/2 kinases and has traditionally been described as an antitumoral mechanism due to its interference between YAP/TAZ and their proliferative and prosurvival target genes [171]. Aberrant YAP/TAZ activity would initially point to potential alterations in Hippo pathway components, which seems unlikely due to the low mutational burden found in human malignancies [172]. However, YAP1 has been described as a target for a rather frequent amplification of locus 11q22 [9,173], particularly in HPV-negative HNSCCs [174]. Other common alterations include upstream regulator and tumor suppressor cadherin *FAT1*, showing deletions in approximately 30% of HNSCCs, contributing to unrestrained YAP activity [16]. 

Other additional regulatory mechanisms regarding YAP/TAZ upregulation and escape from cytoplasm arrest are those related to mechanotransduction, cell–cell and cell–ECM interaction. YAP/TAZ are activated in the presence of mechanical stimuli such as high ECM stiffness, which is a hallmark of HNSCCs [175]. Increasingly fibrotic matrices promote internal modifications in the cytoskeletal and adhesive organization of tumor cells, allowing for YAP/TAZ release and nuclear translocation. As mentioned in previous sections, ECM rigidity is closely related to CAF infiltration and secretion of ECM components. In fact, YAP/TAZ activation also occurs in CAFs, which is necessary for CAF contractility and ECM deposition [176,177].

YAP/TAZ pathway has been linked to multi-drug resistance among a variety of cancers, and is considered of interest in HNSCCs due to common genomic alterations affecting YAP/TAZ activity and regulation, as well as their characteristic fibrotic TME [178]. YAP and TAZ, respectively, have been found to mediate cisplatin resistance in oral and nasopharyngeal SCCs [179]. In OSCCs, the induction of cisplatin resistance correlated with an increase in YAP protein levels, while its silencing sensitized cells to cisplatin. In nasopharyngeal SCCs, cisplatin-resistant cells had an increased EMT phenotype and a gain in invasive capacity, partially dependent on increased TAZ expression in resistant cells [180]. Nevertheless, one of the most important limitations of these studies is that they were performed in 2D culture conditions, where the function of cell–cell interactions and the ECM was omitted.

CSCs or cells with EMT-induced CSC properties are the cornerstone of multidrug resistance. YAP/TAZ activation has been suggested as a possible mechanism through which CSC or CSC-like cell populations are maintained and enriched throughout various malignancies [181]. Studies suggest the acquisition of CSC-like properties through diverse mechanisms; from direct binding of YAP-TEAD-SOX9 in esophageal cancer [182] to YAP-mediated IL-6 secretion [183], which was linked to BMI1 expression in the previous CSC section. Although consistent YAP/TAZ upregulation correlates with CSC enriched tumors, the exact mechanisms through which the Hippo pathway participates are not yet clear.

An interesting aspect to consider is that YAP/TAZ transcriptional co-activators may bind different TFs. This brings about the possibility of YAP and/or TAZ having specific roles, target genes, etc., depending on driver mutations, specific anatomic locations of HNSCCs or the composition of the TME. More detailed analysis will be required to answer these questions. This would explain contradictory results demonstrating that under AKT signaling, YAP acts as a tumor suppressor, inhibiting cell survival and decreasing cisplatin resistance [184].

### 7.2. Survival Pathways: Regulation of STAT3 Pathway

Secretion of IL-6 and other cytokines including IL-11 function by promoting stemness and survival cues to cancer cells and subsequent persistence after cisplatin treatment. For instance, IL-6 binds to its receptor IL-6R-alpha and co-receptor gp130, activating JAK/STAT signaling. Other GF receptors such as EGFR or VEGFR can also trigger this pathway. The signal transducers and activators of transcription (STAT) family is well known to mediate cell survival (stabilizing Bcl-2 and Survivin), proliferation (through c-MYC and CyclinD1), invasion (MMP-9 secretion and EMT induction) and angiogenesis (through VEGF), and its function in HNSCC has been extensively documented [185]. Since few mutations in this pathway are found in HNSCCs, its activation is mainly thought to be caused by upstream cytokines or GFs. STAT3 activation together with NF-κB seems to be more common in HPV-negative HNSCCs [186]. STAT pathway activation is also more prominent in cisplatin-resistant HNSCC models and its inhibition using JAK inhibitors increases cisplatin sensitivity [187,188]. Multiple described mechanisms of cisplatin resistance such as AKR1C1 [189], Rab18 [190,191], and STOML2 [192] rely on the activation of STAT3. In addition, cytokines such as IL6 or IL23 [193] and other molecules secreted by cancer cells or the TME [194] also activate JAK/STAT. Due to the evident implication of the IL6/JAK/STAT pathway in HNSCC progression and therapy resistance, the development of IL-6R-alpha (tocilizumab) and JAK1/2 inhibitors (AZD1480) is extremely promising for the treatment of both primary and cisplatin-refractory HNSCCs, although its clinical benefits remain to be proven.

### 7.3. Notch Pathway

The function of the NOTCH pathway in the initiation, progression, and therapy resistance in HNSCCs remains controversial [15]. While the NOTCH pathway drives differentiation and stratification of skin and esophageal keratinocytes, it drives B and T cell lineage fate decision in the lymphocyte lineage. In cancer, NOTCH gain of function mutations initiate T-cell acute lymphoblastic leukemia, while NOTCH loss of function mutations lead to clonal expansion in the skin and esophageal epitheliums. In HNSCCs, the majority of NOTCH mutations are loss of function, and so NOTCH was considered a tumor suppressor in HNSCCs. However, wild-type NOTCH tumors have an increased expression of the NOTCH target genes HES1 and HEY1, indicating an activation of the pathway. Gu et al. found that NOTCH1 expression was significantly related to cisplatin resistance and that a gamma secretase inhibitor blocking parts of the NOTCH pathway showed a synergistic anticancer effect with cisplatin [187]. NOTCH3 signaling inhibition showed increased cisplatin sensitivity in EBV-associated nasopharyngeal SCCs [195]. Fukusumi et al. showed that NOTCH4 expression specifically correlated with its target HEY1 in patient samples while also driving EMT, stemness and resistance to cisplatin in in vitro assays [196]. It is likely that the roles of the NOTCH pathway will depend on the stage of the tumor, as early-inactivating mutations support tumor initiation. However, if tumors initiate under other genetic and non-genetic events and preserve NOTCH activity, it may likely function as an oncogenic pathway to support aggressive behavior and drug resistance. 

### 7.4. Autophagy

Autophagy is upregulated in various types of cancer, including HNSCC, and has sparked interest as a novel pro-survival mechanism. Canonically employed by the cell to maintain homeostasis, cell survival, and metabolism, intracellular components and organelles undergo degradation within a physiological context. Recent research has pointed towards autophagy as a key player in ECM secretion, mediating transport of well-known tumorigenic mediator molecules (IL-1B, IL-6, IL-8). CAFs showed higher levels of basal autophagy when compared to normal fibroblasts. Inhibition of autophagy with hydroxychloroquine mitigated HNSCC progression and invasive potential in vitro, having a synergistic effect in combination therapy with and RT [197,198]. Higher amounts of LC3-positive autophagosomes have been found both in 4-NQO-induced murine HNSCC models and patient samples [199]. HNSCCs secrete factors such as basic-FGF, which binds to FGFR and activates SOX2 transcription via STAT3 in CAFs, inhibiting mTOR and its downstream effectors. mTOR is a known autophagy suppressor, so releasing this repressive effect could explain the high autophagic activity described in CAFs, which mediates paracrine secretion of tumor-promoting cytokines IL-6 and IL-8, associated with therapeutic resistance. These molecules further maintain a tumorigenic CAF state through an autocrine positive feedback loop [197]. In tongue SCCs, tumor cells co-cultured with CAFs displayed higher autophagic flux through upregulation of key autophagy regulators beclin-1 and LC3-II, and therefore higher viability during cisplatin treatment. This was reversed with autophagy inhibitor chloroquine, which was found to be synergistic with cisplatin [200]. This augmented evasion of apoptosis after cisplatin treatment may drive cell fate towards an increase in autophagy-driven survival mechanisms.

## 8. Conclusions and Future Perspectives

Over the years, it has become clear that CSCs play a critical role in cancer resistance against multiple treatments, especially chemotherapy. HNSCCs are a group of heterogenous cancers that originate from epithelial tissues maintained by specific adult SC populations. Importantly, each adult SC population contain unique characteristics [72] that will be inherited by CSCs once a tumor arises. For example, CSCs from a tongue SCC may be different from one from the nasal cavity or one from the larynx, but a comprehensive analysis to identify these characteristics is yet to be performed. In the same vein, TMEs are notoriously different depending on the tissue of origin. Macrophages, T-cell infiltration, and the insults that shape the immune system, as well as the microbiome, are location-specific. Nevertheless, much deeper studies will be required to understand these differences. Key information may lie within these differences that are relevant to designing more efficient combinatorial treatments specific for each HNSCC location. 

On the other hand, cisplatin resistance mechanisms such as tumor heterogeneity, the presence of various tumor drivers, and cell plasticity have emerged as potential targets to design tailored combinatorial treatments (summarized in Figure 5). Some therapeutic avenues, such as the inhibition of NRF2, blocking of IL-6 function or epigenetic inhibitors (such as HDAD inhibitors) hold promises. Other approaches, such as inhibitors of TGFβ (that could block EMT and immunosuppression of T cells) or macrophage-blocking antibodies, are in the investigational phases. Drugs that, when administered in combination with cisplatin, provoke a synthetic lethality effect on tumor cells (such as the co-inhibition of NRF2 to cause ferroptosis) could allow for lower cisplatin therapeutic doses, preventing its undesirable toxic effects. Other novel but complex pathways, such as the YAP/TAZ pathway and/or autophagy, that are activated by multiple stimuli and play a role in a variety of tumor cell populations are in need of much more investigation to understand the potential systemic effects that their inhibition could cause.

Deeper understanding of the cisplatin mechanisms of action, the identification of synergistic treatments, and the definition of novel biomarkers allowing the prediction of therapy response will be critical to developing alternative regimens and stratifying patients to optimize cisplatin treatment. In addition, the use of cisplatin as a neoadjuvant in combination with numerous immunotherapies targeting the TME is opening up a brand new field in the management of HNSCC patients, although their clinical benefit remains to be revealed.

## Figures and Tables

**Figure 1 cells-11-00561-f001:**
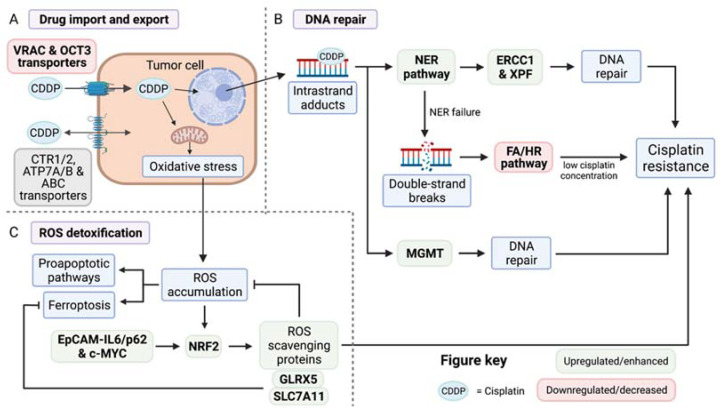
Classical mechanisms of cisplatin resistance in HNSCC. (**A**) Alterations in cellular import and export of cisplatin. Downregulation of VRAC and OCT3 transporters reduces cisplatin cytotoxicity by decreasing its intracellular concentration. While CTR1/2, ATP7A/B and ABC transporters are promising candidates in cisplatin resistance, their functional role in HNSCC is still unknown. (**B**) Activation of DNA repair pathways. Upregulation of NER pathway components ERCC1 and XPF, enhances the removal of DNA intra-strand adducts caused by cisplatin. If NER fails, it will produce DNA double-strand breaks. In this case, a deficit in the FA/HR pathway, which carries out the repair of this type of DNA damage, is related to cisplatin resistance but only at low cisplatin levels. On the other hand, MGMT expression increases the removal of cisplatin-adducts by directly binding to them. (**C**) Enhanced oxidative stress management. Cisplatin induces ROS accumulation, which is counteracted by the overexpression of NRF2. NRF2 upregulates the expression of ROS scavenging proteins, promoting cell survival and, thus, cisplatin resistance. Among these proteins, GLRX5 and SLC7A11 are specifically mentioned, as they are involved in ferroptosis inhibition. Furthermore, NRF2 expression is controlled by EpCAM-IL6/p62 and c-MYC expression. Altogether, these three classical mechanisms converge in cisplatin resistance. Created with BioRender.com (accessed on 1 February 2022).

**Figure 2 cells-11-00561-f002:**
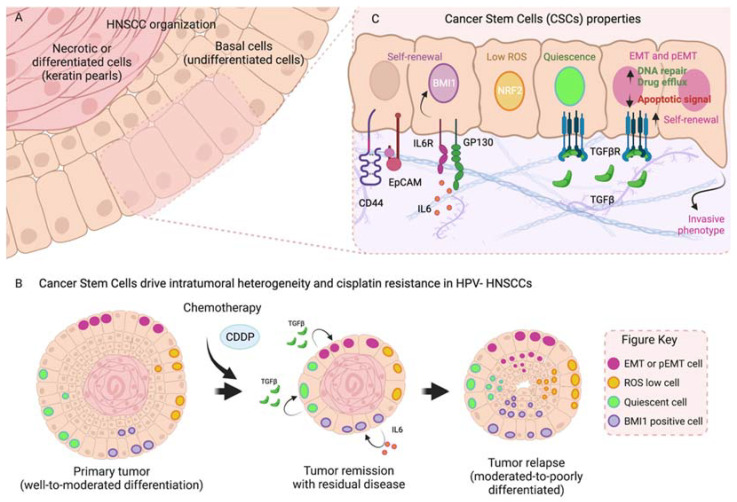
HNSCC CSCs’ characteristics and their role in cisplatin resistance and tumor relapse. (**A**) HNSCCs resemble the architecture of an epithelial tissue, with a basal layer enriched in proliferative cells, potentially containing CSCs. Basal cells give rise to differentiated cells that ultimately form terminally differentiated keratin pearls or necrotic regions. (**B**) HNSCCs present intra-tumoral heterogeneity, containing CSCs pools with different properties, which increases the chances of persisting after cisplatin treatment. CSCs surviving after chemotherapy could repopulate the affected zone causing tumor relapses that are usually poorly differentiated. (**C**) CSCs’ properties include increased self-renewal promoted by IL-6-induced BMI1 expression, enhanced ROS detoxification driven by NRF2 overexpression and the induction of quiescence or mesenchymal programs through TGFβ. Created with BioRender.com.

**Figure 3 cells-11-00561-f003:**
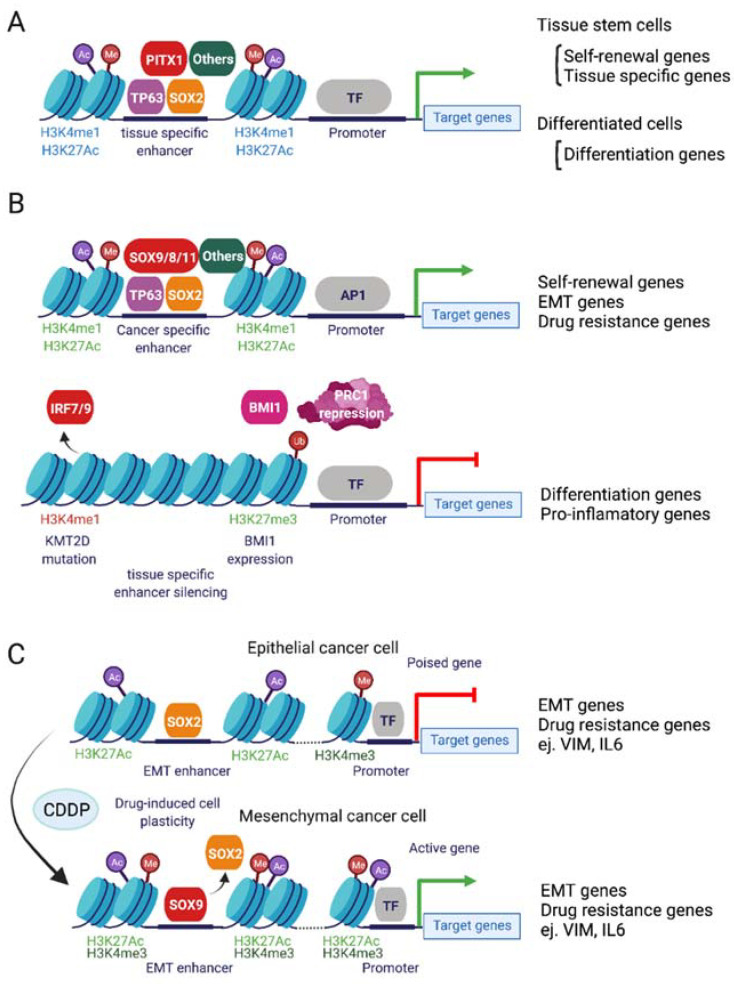
Epigenetic reprogramming in cisplatin-resistant HNSCC. (**A**) In physiological conditions, tissue homeostasis is supported by the expression of the TFs PITX1, TP63 and SOX2. (**B**) In tumor initiation events, two different landscapes can occur. On one hand, overexpression of oncogenic TFs promotes chromatin accessibility to cancer-specific enhancers, which results in self-renewal, EMT and drug resistance genes increase. On the other hand, the loss of active marks (H3K4me1) and the gain of repressive marks (H3K27me3) near tissue-specific enhancers lead to the silencing of tissue differentiation and pro-inflammatory genes. (**C**) Tumors adapt to cisplatin by replacing SOX2 with SOX9 in EMT enhancers as well as the deposition of the H3K27Ac mark, allowing the expression of mesenchymal and drug resistance genes. Created with BioRender.com.

**Figure 4 cells-11-00561-f004:**
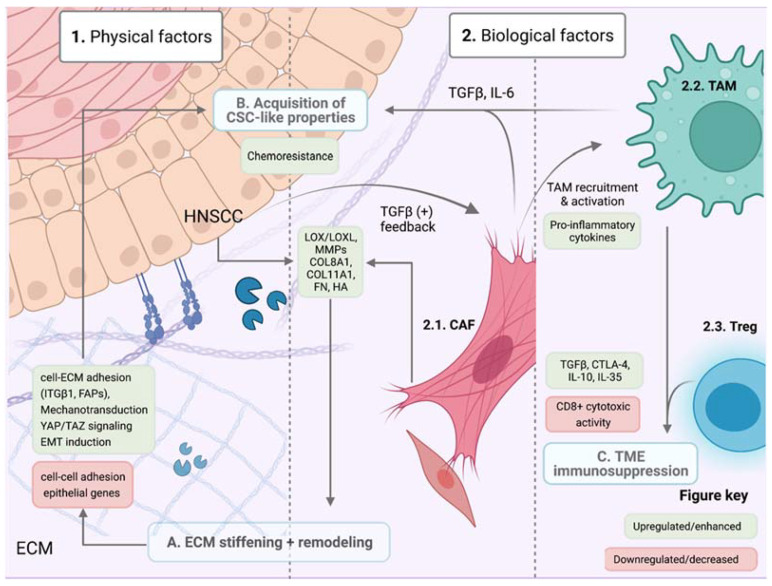
TME involvement in chemoresistance development. Tumor cell-extrinsic influence on therapy response can be divided into physical (1) and biological (2) factors. HNSCC cells can recruit surrounding fibroblasts (middle section) through TGFβ signaling, which induces an activation of the CAF phenotype (2.1). CAFs then further produce TGFβ establishing a positive feedback loop between cancer and stromal cells. TGFβ pathway activation promotes the deposition of matrix proteins (COL8A1, COL11A1, FN, HA) and remodeling enzymes (LOX/LOXL, MMPs), which help to reshape and shift the ECM composition. As a result, the matrix becomes stiffer (**A**), promoting cytoskeletal reorganization within HNSCC cells and thus cell–ECM interactions through increased FAP and integrin expression. Increased cell–ECM adhesion favors the activation of integrin-dependent signaling and mechanosensitive pathways (such as YAP/TAZ), which downregulate cell–cell interactions, enhancing motility and the acquisition of an EMT-resistant phenotype (**B**). CAFs can also secrete inflammatory cytokines such as IL-6, which can both induce CSC-like properties in HNSCCs (**B**) and recruit other TME populations, such as immune cells. TAMs (2.2) become active in response to proinflammatory cytokines to further secrete TGFβ, supporting the activation of this pathway. Together with infiltrated T regulatory lymphocytes (2.3), they help to maintain an immunosuppressed tumoral microenvironment which is likely to evade immunotherapy (**C**). Created with BioRender.com.

**Figure 5 cells-11-00561-f005:**
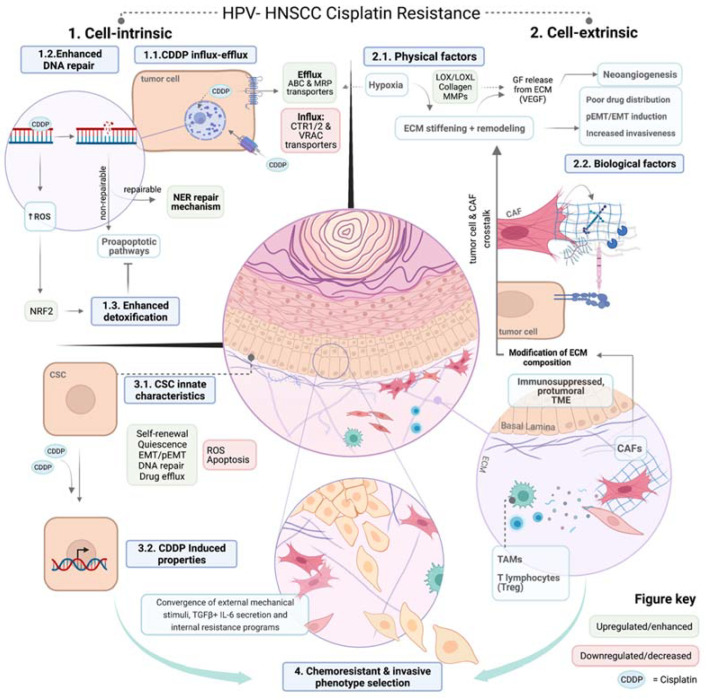
Cell-intrinsic (1) and -extrinsic (2) cisplatin resistance mechanisms converge to promote aggressive, invasive and chemoresistant phenotype selection in HPV-negative HNSCC (4). Classic drug resistance mechanisms ((1), also in Figure 1) include efficient detoxification through deregulated transporter expression (1.1), enhanced NER DNA repair (1.2) and increased antioxidant mechanisms via the upregulation of NRF2 (1.3). CSCs displaying this innate chemoresistant behavior ((3.1), also in Figure 2C) may additionally intensify or boost these properties in response to cisplatin treatment (3.2), contributing to resistant clone selection which proliferates and gives way to relapses ((4), also in Figure 2B). Cell-extrinsic factors (2) can be divided into physical/mechanical factors (2.1) and biological components of the TME (2.2). As tumors grow, shear pressure makes the distribution of chemotherapy inefficient and nutrient supply becomes insufficient, triggering hypoxia response programs, which include the upregulation of efflux transporters (1.1) and ECM remodeling components (LOX/LOXL enzymes, collagens and MMPs), which cleave and restructure the ECM, releasing embedded factors such as VEGF, involved in neoangiogenesis. Other factors contributing to ECM remodeling are non-tumoral cells within the TME (2.2), which establish an intricate crosstalk with the neighboring tumor. Important interactions include bidirectional TGFβ and IL-6 feedback loops between tumoral and non-tumoral populations (CAFs, TAMs and Tregs), which induce ECM protein deposition and stiffening (Figure 4). This modifies the adhesive distribution within tumoral cells, increasing cell–ECM interactions and triggering mechanosensitive pathways. These ultimately promote EMT expression programs, favoring invasive and resistant behavior (4). Created with BioRender.com.

**Table 2 cells-11-00561-t002:** FDA-approved and investigational drugs available for HNSCC treatment.

Approved	Experimental (Ongoing Clinical Trials)
**First-line treatment in early/advanced stages** Surgical excision and Radiotherap Chemotherapy (platin-based (cis/carboplatin), 5-FU, cetuximab)	**Small molecules** ***HRAS^mut^* Farnesyltransferase inhibitor:** Tipifarnib [13] ***CDK4/6*****inhibitor:** Palbociclib [15] **Tyrosine-kinase inhibitors (TKIs):** Targeting EGFR: Afatinib *, Erlotinib, Gefitinib, Lapatinib ** Targeting VEGFR: Sunitinib Targeting MET kinase: Tivantinib [14]
**First-line treatment for recurrent, unresectable and metastatic tumors** **Chemotherapy** Monotherapy: Cis/carboplatin, docetaxel, paclitaxel, 5-FU, cetuximab, gemcitabine [19,20] Combinatory therapy: Cis/carboplatin + (docetaxel, paclitaxel, 5-FU, cetuximab or gemcitabine) Cis/carboplatin + cetuximab + (docetaxel, paclitaxel, 5-FU or gemcitabine) [21,22,23,24,25,26,27] **Targeted Immunotherapy (mAb):** Nivolumab, Pembrolizumab (anti-PD-1) [28,29,30,31]	

Current FDA-approved first-line treatment for HNSCC patients (upper left corner) include surgical excision, traditional cytotoxic chemotherapy, as well as molecularly targeted agents such as cetuximab. Other newly authorized monoclonal antibodies (mAb) such as nivolumab and pembrolizumab are also implemented, mostly in treatment refractory cases. Combined or multimodal treatments between these therapies seem to provide greater benefit compared to monotherapy. Emerging investigational therapies (right box) aim to satisfy the current medical need for better clinical outcomes, focusing on small molecule-based drugs. Most of them are still going through clinical phases and not yet marketed. (*) = ongoing trials: NCT01783587, NCT03088059, NCT02979977; (**) = ongoing trials: NCT01044433, NCT01711658, NCT01612.

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
