# Peer review of "Mechanisms of Cisplatin Resistance in HPV Negative Head and Neck Squamous Cell Carcinomas"

_cells, 2022, doi:10.3390/cells11030561_

Round 1
Reviewer 1 Report
Comment:
The current version of the manuscript [[Cells] Manuscript ID: cells-1566207] did not reach the standard of expert review articles for [Cells], a journal with impact factor 6.6. The clarity should be enhanced by addressing the following specific comments.
Specific comments:
- Line 12: "In this review, we summarize different mechanisms involved in" to Line 23 "interplay between extracellular matrix content as well as immune and non- "lack the logical flow of importance (their theme).
- Lines 50 – 127 should have come with tables to increase the clarity.
- Lines 119-127: "The introduction should briefly place the study in a broad context and highlight why it is important. It should define the purpose of the work and its significance. The current state of the research field should be carefully reviewed, and key publications cited. Please highlight controversial and diverging hypotheses when necessary. Finally, briefly mention the main aim of the work and highlight the principal conclusions. As far as possible, please keep the introduction comprehensible to scientists outside your particular field of 124 research.
References should be numbered in order of appearance and indicated by a numeral or numerals in square brackets—e.g., [1] or [2,3], or [4–6]. See the end of the document for further details on references."
This paragraph appears directly to be taken from a peer-review report. It does not belong in an expert review manuscript. It is problematic with this manuscript.
- Lines 158 – 499: The authors should construct their own schematic diagrams to govern their thought lines.
- Lines 550-512: the section seems too shallow to reflect on the TME literature to provide insight to the reader.
- Lines 670-717: they piled up the facts; however, they did not extrapolate the themes.
- Lines 736-737: "particularly in HPV" – little is tied into HPV effects.
- Line 778: They freely exchange the concept of cancer cells and CSCs.
- Lines 841-872: In conclusion, they wrote like Introduction all over again.
- Line 875: Figure legend should be Fig 1.
- Lines 875-876: "Cell-intrinsic and extrinsic cisplatin resistance mechanisms converge to promote aggressive, highly invasive and chemoresistant tumoral phenotypes in HPV-negative HNSCC" – the sentence is not logical, as the process can go either way.
- Line 880: "These helps maintain cisplatin at subtherapeutic" Grammar error.
- Lines 884-885: "undergo EMT/pEMT promoting gain and loss of mesenchymal and epithelia" should be given citations.
- Line 896: "triggering mechanotransduction through YAP/TAZ" should be marked with citations.
- Not sufficient discussion on Cancer Subclones Derived from the Patient's Head and Neck Squamous Cell Carcinoma Tumor Stem Cells for the Screening of Personalized Antitumor Immunotherapy and Chemotherapy. Stem Cell Research & Therapeutics (ISSN: 2474-4646). 2019 February; 3(1):116-121. doi: 10.13140/RG.2.2.32157.97769.
- Multiple fine points should be considered in the sentence structure, punctuations, and style of English.
Reviewer 2 Report
This is a thorough review about head & neck SC carcinoma and resistance covering well known aspects and novel topics such as epigenetic regulation of cell plasticity. It goes far beyond cis-platinum resistance mechanisms. I have some minor comments:
- Given the amount of data covered, some figures or tables would be desirable. For example, with the molecular alterations in HNSCCs tumors.
- Lines 119 to 127 needs to be deleted.
- Please check for minor mistakes, like double use of also in line 200.
- Avoid quoted words like "activated".
Reviewer 3 Report
Review of manuscript by Griso et al. entitled “Mechanisms of cisplatin resistance in head and neck squamous cell carcinomas”. The review has strengths and is focused on very important problem in the field.
There are following concerns:
- There is no flow in the reading in lot of places. Paragraphs are not connected well. Authors have to connect each information before jumping to another topic/point. This will help readers and create interest in reading.
- Please provide a table highlighting the pathways/molecules involved in resistance.
- line #88, it should be “review” not “revision”. Please check.
- lines 119-127, the whole paragraph is not required. Please check.
- Schematic Figure: Looks good but difficult to follow. Physical factors are number 1 then biological factors are given number 2. Other headings should be numbered. What comes under these factors? Figure should be mentioned in the text.
Round 2
Reviewer 1 Report
accepted.